# First Report of *Fusarium andiyazi* Presence in Portuguese Maize Kernels

Daniela Simões [1], Eugénio Diogo [1,2] and Eugénia de Andrade [1,3,*]

1   National Institute for Agricultural and Veterinary Research (INIAV), I.P., Unidade Estratégica de Sistemas Agrários e Florestais e Sanidade Vegetal, 2780-159 Oeiras, Portugal; daniela.simoes@iniav.pt (D.S.); eugenio.diogo@iniav.pt (E.D.)
2   BioISI—Biosystems & Integrative Sciences Institute, Faculty of Sciences, University of Lisbon, 1749-016 Lisbon, Portugal
3   GREEN-IT Bioresources for Sustainability, ITQB NOVA, Av. da República, 2780-157 Oeiras, Portugal
*   Correspondence: eugenia.andrade@iniav.pt

**Abstract:** Maize is one of the most important staple crops worldwide. However, it is also very susceptible to fungal infections. Some *Fusarium* species are responsible for causing diseases in maize and producing mycotoxins, contributing for considerable reduction of yield, quality, and profit. In Portugal and other Mediterranean countries with similar climatic conditions, *F. verticillioides* and *F. subglutinans* are the most frequent species infecting maize. *F. proliferatum*, *F. graminearum*, and *F. oxysporum* are only occasionally found. However, the incidence of diseases and the occurrence and levels of fumonisins have increased, which indicate that either the exogenous conditions changed favouring the production of mycotoxins, or other mycotoxigenic species of *Fusarium* are present. Therefore, *Fusarium* species occurrence in field should be monitored. After a survey of *Fusarium* spp. in Ribatejo county, for the first time we report *F. andiyazi* in Portuguese maize crop. This species is pathogenic for maize under similar climatic conditions, and mycotoxigenic, which means a double concern to the farmers if its presence on maize increases. This report highlights the importance of surveying and monitoring local fungal diversity on maize to enable stakeholders of the maize-chain production to respond to threats. Further studies to prevent *Fusarium* spreading in maize need to ,.

**Keywords:** *Zea mays*; *Fusarium* spp.; mycotoxins; Portugal

## 1. Introduction

Maize is one of the most important staple crops used in human diets and is among the most important livestock feed sources worldwide [1,2]. Between 2016 and 2018, it reached a global production averaging 1136 million tonnes. Worldwide, 12% of the production is for human consumptions and 56% is used as feed [3]. In Portugal, during the same period, maize was a crucial crop for the national economy, having been the cereal with the highest production of grain (with over 700,000 tonnes per year) [4]. However, maize crops are very susceptible to fungal infections, which can be caused by a large range of fungal species, including mycotoxigenic species. These fungal infections are responsible for significant losses in yield and quality [1], being a big concern to farmers and consumers. *Fusarium* is one of the major pathogenic genera affecting maize crops, mainly due to *Fusarium fujikuroi* complex species, as *F. verticillioides*, *F. subglutinans*, *F. proliferatum*, and *Fusarium graminearum* [5]. Although infections by *Fusarium* species may provoke a defective establishment of the crop and, consequently, lower yields. The most important concern is the highly probable contamination of kernels with mycotoxins, such as fumonisins [6–11]. The consumption of mycotoxins-contaminated food or feed have an impact on human and animal health resulting in acute or chronic consequences such as carcinogenic, teratogenic, immunosuppressive, or estrogenic issues [12]. From 2018 to 2020, a study to evaluate and monitor the presence of mycotoxins in maize grown in the major maize production

region of Ribatejo province, the Municipality of Golegã (39°24′ N, 8°29′ W) located in the Tagus Valley region (Portugal), has been performed and showed a high prevalence of fumonisins [8]. Thus, the aim of this study was to screen maize grains field-sampled, from one of the most important productive regions in Portugal, for *Fusarium* species. Exploring early detection and control of mycotoxigenic *Fusarium* species is a crucial step to decrease mycotoxins incidence and increase the safety of maize-based food and feed.

## 2. Materials and Methods

### 2.1. Growing Conditions and Sampling

Out of the 2018 campaign, a total of 17 maize samples were collected from 7 plots in 3 different farms and analysed to identify the different *Fusarium* species present on. Composite samples of 5 Kg of maize freshly harvested were collected in the field for the detection of both, mycotoxins and the associated fungi. Subsamples of 0.5 Kg were received in the Mycology and Molecular Biology Labs.

The soils of these three farms, generally classified as Calcic Cambisols [13], are presenting a texture mostly sandy loam. They are not homogenous, including patches of sandy, loamy, and clayish soil.

In 2018, the average temperature was 17.2 °C, varying between 11.2 °C and 23.2 °C. The total annual precipitation was 713.3 mm, mostly between March and April (totalizing 299 mm (41.9%) only in these two months). During the maize development period, from May to October, the average temperature variated between 17.8 °C and 26 °C, presenting a minimal temperature of 11.4 °C (May) and a maximum temperature of 34.7 °C (August). The total precipitation in this period was in average 21.3 mm, variating between 0 mm (July and August) and 50.6 mm (June) [14].

### 2.2. Isolation and Morphological Characterization

The fungal analysis was carried out following the protocol described by Carbas and co-workers [8], plating 50 pre-disinfected grains of each sample in Malachite Green Agar. All of the monosporic isolates were incubated at 27 °C and observed after 7 and 10 days of incubation. The isolates were identified by observation of their macro and microscopic characteristics in PDA and CLA media following Leslie and Summerell [5]. Two isolates morphologically similar to *Fusarium andiyazi* were observed. The two *F. andiyazi* monosporic isolates transferred onto PDA presented white and violet powdery mycelium, with violet pigmentation in the agar. On CLA (carnation leaf agar), long chains of ovoid-clavated microconidia from long monophialides and singly pseudochlamydospores were observed.

### 2.3. Molecular Characterization

The identifications of *Fusarium* species isolates were confirmed by sequencing the *Translation Elongation Factor* 1-α (*TEF*) gene fragment whenever necessary. As *F. andiyazi* was never observed in Portugal, the molecular confirmation of the identity of these isolates was carried out by sequencing fragments of both *TEF* and *β-tubulin* genes.

DNA was extracted following the Chelex method [15,16], using Chelex® 100 Chelating Resin, analytical grade, 200–400 mesh (Bio-Rad Laboratories, Hercules, CA, USA). The genomic *TEF* fragment was amplified using the primers EF1-728F and EF1-986R [17]. Polymerase chain reaction (PCR) conditions were as described in Carbas et al., 2021 [8]. To amplify the *β-tubulin* fragment, primers Bt1a and Bt1b sequences were obtained from Glass and Donaldson [18]. PCR for *β-tubulin* fragment were performed in 25 μL-mixture containing 2.5 μL of 5x enzyme buffer, 0.5 μL of each primer (10 μM), 0.2 μL of dNTPs (10 mM), 1.25 μL of MgCl$_2$ (25 mM), 0.2 μL of GoTaq®G2Flexi DNA polymerase (5 U/μL) (Promega, Walldorf, Germany), 17.85 μL of ultrapure water and 2 μL of DNA (independently of the concentration in ng/μL). PCR amplification consisted of the following cycle parameters: an initial step at 94 °C for 2 min, followed by 30 cycles at 94 °C for 60 s, 48 °C for 60 s, and 72 °C for 60 s. A final extension step at 72 °C for 5 min was added. All the amplification reactions were performed in a Biometra Tone thermocycler (Analytik Jena GmbH, Jena,

Germany). Five microliters of each PCR reaction product were analysed in a 2% agarose gel stained with GelRed. The products were visualized and photographed under UV light (254–365 nm), (GboxHR, Syngene, Cambridge, UK) and the remaining PCR reaction was purified with illustra™ ExoProStar™, (GE Healthcare Life Sciences, Buckinghamshire, UK) and sequenced in the Unit for Plant Breeding, INIAV, Oeiras. The isolates were sequenced with the same primers as those used for the amplification and in both forward and reverse directions. Their sequences were compared with the sequences deposited in GenBank, NCBI (https://blast.ncbi.nlm.nih.gov/Blast.cgi, accessed on 23 March 2021) and only accepted when with coverage and homology both higher than 98%. The obtained sequences of the two *F. andiyazi* isolates were submitted to the GenBank database, and the correspondent accession numbers are MW689615, MW689616, MW79564, and MW795648.

## 3. Results and Discussion

From the 850 maize grains analysed, 354 *Fusarium* isolates were obtained. The most prevalent species found was *F. verticillioides* (79.7%), followed by *F. subglutinans* (17.5%), and *F. proliferatum* (1.4%). Also *F. graminearum* (0.6%), *F. andiyazi* (0.6%), and *F. oxysporum* (0.3%) were found.

*Fusarium andiyazi*, first described on sorghum in Africa and in the United States [19], was also found on maize in Syria, China, Mexico, and recently in Uganda and Italy [20–24]. This species was reported causing fusarium ear rot in maize grains, with its pathogenicity tested [21,23], and described as mycotoxigenic. *Fusarium andiyazi* can produce Fusaric acid, Fusarin C [25,26], and fumonisins [24]. This means that *F. andiyazi* has the potential to be pathogenic and mycotoxigenic on maize under the favourable conditions. The climatic conditions in Portugal and Italy are similar, thus it is probable that *F. andiyazi* causes Fusarium ear rot disease in maize in both countries. According to Köppen–Geiger Climate Classification, South of Portugal, including Tagus Valley Region, is a country with temperate climate type Csa, such as most of the Italian territory [27]. Further investigation is needed to understand the spatial and temporal dynamics of this species and pathogenicity in Portuguese maize.

## 4. Conclusions

This work reports, for the first time, *Fusarium andiyazi* in Portugal, expanding its known geographical range. This report shows the importance of conducting further prospection studies to increase the knowledge of the maize microbiomes to predict issues and anticipate the research of solutions. Even though the reduced prevalence of *F. andiyazi* in this study, with the evident increasingly climactic changes it is predicted that microbiomes also change, being probable that the prevalence of fungal species will also change. In this way, scientific efforts to understand better the nowadays non-major fungal species ecology, genetics, pathology, and toxigenicity should be encouraged.

**Author Contributions:** Conceptualization: D.S. and E.d.A.; methodology: D.S., E.D. and E.d.A.; investigation: D.S.; data curation: D.S.; writing—original draft preparation: D.S.; writing—review and editing: E.d.A. and E.D.; supervision: E.d.A. and E.D.; project administration: E.d.A.; funding acquisition: E.d.A. All authors have read and agreed to the published version of the manuscript.

**Funding:** This research was funded by the National Funds dedicated to the Rural Development Program through the Operational Group QUALIMILHO—New sustainable integration strategies that guarantee quality and safety in the national maize, PDR2020 n° 101-031295 (2017–2020). GREEN-IT Bioresources for Sustainability, ITQB NOVA, Av. da República, 2780-157 Oeiras, Portugal.

**Informed Consent Statement:** Not applicable.

**Data Availability Statement:** Molecular data generated in this study were deposit in GenBank.

**Conflicts of Interest:** The authors declare no conflict of interest.

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
