# Peer review of "First Report of Fusarium andiyazi Presence in Portuguese Maize Kernels"

_agriculture, doi:10.3390/agriculture12030336_

Round 1
Reviewer 1 Report
Comments and suggestions to authors.
Communication
First Report of Fusarium Andiyazi Presence in Portuguese Maize Kernels
agriculture-1603465
Some comments that can help improve the text:
2 - Title, scientific names must be in cursive and the species names must start with a lowercase.
31 – tons, check abbreviation.
34 – Again, check abbreviation, here is written as t.
39 – Delete the “and” before F. proliferatum.
43 – There is a lack of justification and the objectives that were raised to do this work.
45 – This part is very messy. It begins with a justification rather than clearly explaining the study made, the sampling methods. Two strains that resemble F. andiyazi are then studied by molecular methods (sequencing). The work seems to be a survey of mycotoxigenic and/or pathogenic Fusarium spp. in corn samples and thus, F. andiyazi strains were found in the whole identification of strains, explain better.
46 – 49 - It seems more like a justification than Materials and methods, remove and add it in the Introduction.
49 – What was the criteria for obtaining the samples?
52 – 54 - It is not clear how they determined/supposed that it was this species, explain better.
81 – Looks like Results and discussion.
82 - Here it is explained that 354 strains were obtained, some of them F. andiyazi. The identifications (of those strains for which percentages of colonization are shown) were by the same methods? That is, sequencing of b-tubulin and TEF.
86 – 89 – Does this agree with other descriptions in the cited literature?
96 – “Thus it is probable that F. andiyazi causes disease (Fusarium ear rot) also in Portuguese maize” … No symptoms like those mentioned were found in the survey?
Author Response
Communication
First Report of Fusarium andiyazi Presence in Portuguese Maize Kernels - agriculture-1603465
Referee 1 -
First of all we would like to really thank the comments and the effort made by the reviewer aimed to improve the manuscript. We replied to all comments included point by point
Comment 1- line 2
Title, scientific names must be in cursive and the species names must start with a lowercase.
R: We submitted the manuscript (Ms) as an ordinary word document. We used cursive. We think that the conversion to the journal format changed the way to refer the scientific name. The species name was amended according to the reviewer’s opinion.
Comment 2 - line 31
tons, check abbreviation
R: the word was corrected as required.
Comment 3 - line 34
Again, check abbreviation, here is written as t.
R: the abbreviation was corrected as required.
Comment 4 - line 39
Delete the “and” before F. proliferatum.
R: the correction was made as suggested.
Comment 5 - line 43
There is a lack of justification and the objectives that were raised to do this work.
R: Thank you for this alert. We wrote the communication as a “conventional” first report. The justification was in the abstract. The text was amended according to the reviewer’s opinion.
Comment 6 - line 45
This part is very messy. It begins with a justification rather than clearly explaining the study made, the sampling methods. Two strains that resemble F. andiyazi are then studied by molecular methods (sequencing). The work seems to be a survey of mycotoxigenic and/or pathogenic Fusarium spp. in corn samples and thus, F. andiyazi strains were found in the whole identification of strains, explain better.
R: We appreciate the comment. As justified above, we wrote the Ms as a conventional “first report”. Ttaking the format of the communication, we changed the text, to explain better.
Comment 7 - line 46 – 49
It seems more like a justification than Materials and methods, remove and add it in the Introduction.
R: The text was amended as suggested by the reviewer.
Comment 8 - line 49
What was the criteria for obtaining the samples?
R: On the field, the sampling followed the requests of the team responsible for the detection and identification of mycotoxins. Out of their samples, 0.5 Kg of grains was sent to the mycology and molecular biology labs. We added this information to the text.
Comment 9 - line 52 – 54
It is not clear how they determined/supposed that it was this species, explain better.
R: The morphological identification was made by observation of the macro and microscopic characteristics of the isolates in PDA and CLA, following “The Fusarium Laboratory Manual” by Leslie and Summerell (2006). This information was added to the text.
These isolates presented a very similar morphology to some Fusarium verticillioides isolates already collected, but with singly pseudochlamydospores present. According to Leslie and Summerell (2006), seven species are described as being very similar to F. verticillioides: F. fujikuroi, F. globosum, F. napiforme, F. nygamai, F. pseudonygamai, and F. thapsinum. However, none of them are described as presenting all the observed characteristics simultaneously (purple aerial mycelia and agar pigmentation in PDA; long monophialides with long chains of ovoid-clavated microconidia and singly pseudochlamydospores in CLA), which indicated us that these two isolates were probably F. andiyazi. Therefore, the identifications were confirmed by sequencing TEF and β-tubulin genes.
Comment 10 - line 81
Looks like Results and discussion.
R: the subtitle was amended as suggested by the reviewer.
Comment 11 - line 82
Here it is explained that 354 strains were obtained, some of them F. andiyazi. The identifications (of those strains for which percentages of colonization are shown) were by the same methods? That is, sequencing of b-tubulin and TEF.
R: the number of isolates was too high to use sequencing of two genes in all. The mycologists of the team are experienced senior scientists who do not need to use molecular biology techniques in all situations. All not totally clear identifications were confirmed by the sequencing of a TEF gene fragment. We added this information to the text.
Comment 12 - line 86 – 89
Does this agree with other descriptions in the cited literature?
R: This description agrees with the book that we followed to identify all the isolates (“The Fusarium Laboratory Manual” by Leslie and Summerell (2006)).
Comment 13 - line 96
“Thus it is probable that F. andiyazi causes disease (Fusarium ear rot) also in Portuguese maize” … No symptoms like those mentioned were found in the survey?
R: In order to guarantee the representativeness of the sample, the plated grains were randomly selected, without special attention to their health appearance, so we cannot guarantee that the origin of these isolates. They could be obtained either from healthy or symptomatic grains

Reviewer 2 Report
Communication presents the first occurrence of Fusarium andiyazi in maize kernels in Portugal (in Tagus Valley, in Ribatejo Province), in 2018.
From 17 maize kernels samples, 850 maize grains were analysed and 354 Fusarium isolated were obtained. Fungal species were identified by PCR method, as F. verticillioides, F. subglutinans, and F. proliferatum, also F. graminearum, F. andiyazi and F. oxysporum.
- F. andiyazi presented white and violet powdery mycelium, with violet pigmentation in the potato dextrose agar (PDA), and long chains of ovoid-clavated microconidia from long monophialides and singly pseudochlamydospores on carnation leaf agar (CLA).
I recommend some improvements as follows:
1). Geographic coordinates: Portugal (38°42′ N, 9°11′ W), Ribatejo province (39°14′ N, 8°41′ W), or for the Tagus Valley (40°19′ 6.60’’ N, 1° 41’ 30.59‘’ W)
2). Climate in the Tagus Valley region – according to Köppen–Geiger Climate Classification.
https://www.researchgate.net/figure/Koeppen-Geiger-climate-type-map-of-Europe_fig3_26640584
3) Soil type in the Tagus Valley region.
4). Meteorological parameters in Tagus Valley, in 2018: air temperature and precipitation – annual average and in the critical period for Fusarium spp. attack in maize.
5) L20: … F. andiyazi in Portuguese maize production crop.
6). L46: … mycotoxins in maize kernels grown in the Tagus valley ….
7). L49: … a total of 17 maize grain samples….
8). L85: Growing temperature of each Fusarium isolates.
9). L96: The climatic conditions in Portugal and Italy are similar, thus it is probable that F. andiyazi causes Fusarium earrot disease in maize in both countries.
10). Some conclusions taking into account these points.
Author Response
Communication
First Report of Fusarium andiyazi Presence in Portuguese Maize Kernels - agriculture-1603465
Referee 2
First of all we would like to really thank the comments and the effort made by the reviewer aimed to improve the manuscript. We replied to all comments included point by point
Communication presents the first occurrence of Fusarium andiyazi in maize kernels in Portugal (in Tagus Valley, in Ribatejo Province), in 2018.
From 17 maize kernels samples, 850 maize grains were analysed and 354 Fusarium isolated were obtained. Fungal species were identified by PCR method, as F. verticillioides, F. subglutinans, and F. proliferatum, also F. graminearum, F. andiyazi and F. oxysporum.
- F. andiyazi presented white and violet powdery mycelium, with violet pigmentation in the potato dextrose agar (PDA), and long chains of ovoid-clavated microconidia from long monophialides and singly pseudochlamydospores on carnation leaf agar (CLA).
I recommend some improvements as follows:
Comment 1)
Geographic coordinates: Portugal (38°42′ N, 9°11′ W), Ribatejo province (39°14′ N, 8°41′ W), or for the Tagus Valley (40°19′ 6.60’’ N, 1° 41’ 30.59‘’ W)
R: We implemented the suggestion of the referee with the introduction of the coordinates of the municipality.
Comment 2)
Climate in the Tagus Valley region – according to Köppen–Geiger Climate Classification.
https://www.researchgate.net/figure/Koeppen-Geiger-climate-type-map-of-Europe_fig3_26640584
R: We implemented the suggestion of the referee with the introduction of the climatic type of Tagus Valley Region.
Comment 3)
Soil type in the Tagus Valley region.
R: Our survey was done in three farms. The soil was not homogeneous. It was clear the presence of patches of sandy, loamy, and clayish soil, being the sandy loam texture the most frequent. The text was amended according to the recommendation of the reviewer.
Comment 4)
Meteorological parameters in Tagus Valley, in 2018: air temperature and precipitation – annual average and in the critical period for Fusarium spp. attack in maize.
R: The annual average air temperature and precipitation were added to the text, such as the air temperature and precipitation during the months of maize development. As the sampling was only made at the maize harvesting, we do not know exactly when the critical period occurred. Previously to this work, we have studied the progression of Fusarium species inside plants of different varieties (not yet published; under preparation). We know that Fusarium sp. can be detected inside the plant within few weeks after sowing, either when sowing is at the beginning of April or later in May. It can also be detected throughout the life maize life cycle. Thus, as infections may be soil born or airborne, the environmental conditions have an important role in Fusarium infection and contamination of maize during all its life cycle.
Comment 5)
L20: … F. andiyazi in Portuguese maize production crop.
R: the text was amended according to the reviewer’s correction.
Comment 6)
L46: … mycotoxins in maize kernels grown in the Tagus valley ….
R: the text was amended according to the reviewer’s correction.
Comment 7)
L49: … a total of 17 maize grain samples….
R: the text was amended according to the reviewer’s correction.
Comment 8)
L85: Growing temperature of each Fusarium isolates.
R: All Fusarium isolates grew at the same temperature (27ºC). We added this information to the text.
Comment 9)
L96: The climatic conditions in Portugal and Italy are similar, thus it is probable that F. andiyazi causes Fusarium earrot disease in maize in both countries.
R: the text was amended according to the reviewer’s correction.
Comment 10)
Some conclusions taking into account these points.
R: We implemented the suggestion of the referee.

Round 2
Reviewer 1 Report
Review for agriculture-1603465
Communication
First report of Fusarium andiyazi presence in Portuguese maize kernels
Daniela Simões , Eugénio Diogo , Eugénia de Andrade
The suggestions provided in the previous review have been correctly answered and/or corrected. In addition, valuable information has been incorporated when requested or advised.
Therefore, its publication is recommended in the present format.
This article can be an extra contribution to the knowledge about the distribution and importance of toxicogenic species of Fusarium.